# Psychological Features of Fibromyalgia in the Psychological Health Services

**DOI:** 10.3390/bs14111016

**Published:** 2024-11-01

**Authors:** Gabriela Rios Andreghetti, Sonia Montemurro, Luca Rizzi, Laura Casetta, Marcello Passarelli, Sara Mondini, Diego Rocco

**Affiliations:** 1Department of Philosophy, Sociology, Education and Applied Psychology (FISPPA), University of Padua, 35122 Padua, Italy; gabriela.riosandreghetti@studenti.unipd.it (G.R.A.); sonia.montemurro@unipd.it (S.M.); 2Associazione Centro di Psicologia e Psicoterapia Funzionale, Istituto SIF di Padova, 35138 Padua, Italy; luca.rizzi@ymail.com (L.R.); laura.casetta@yahoo.it (L.C.); 3Institute of Educational Technology, National Research Council, 16149 Genoa, Italy; marcello.passarelli@cnr.it; 4Department of Developmental Psychology and Socialization (DPSS), University of Padua, 35131 Padua, Italy; diego.rocco@unipd.it; 5Human Inspired Technology—Research Centre HIT, University of Padua, 35122 Padua, Italy; 6Servizi Clinici Universitari Psicologici (SCUP)—Centro di Ateneo, University of Padua, 35131 Padua, Italy; 7IRCCS San Camillo Hospital, 30126 Venice, Italy

**Keywords:** fibromyalgia, psychological assessment, mental health

## Abstract

Patients with health pathologies may exhibit psychological features in addition to medical symptomatology. A sample of 76 Italian women with an age range between 23 and 78 years old (mean = 50.22 ± 10.47 years) diagnosed with fibromyalgia (a disorder characterized by widespread musculoskeletal pain accompanied by fatigue, sleep, cognitive and mood issues) was examined to identify typical characteristics of their psychological profile. All patients were administered a series of questionnaires, to assess avoidance of physical touch by others; anxiety; depression; risk of developing psychotic disorders; self-criticism and perfectionism; acceptance of chronic pain; general impact of the pathology on their lives, and the predisposition to experience positive emotional states. The scores resulting from this assessment were evaluated against the normative data. Patients with fibromyalgia showed psychological dysfunction in most of the scales administered, with a significantly higher disposition to experience compassion towards others (*t* = 5.94, df = 75, *p* < 0.001). A higher risk of psychosis was related with higher levels of depression (*B* = 0.49, *t* = 0.20, *p* = 0.015), poor involvement in daily activities (*B* = 0.41, *t* = 0.13, *p* = 0.002), touch avoidance (especially towards strangers, *B* = −0.20, *t* = 0.14, *p* = 0.01), and overall poor quality of life (*B* = 0.40, *t* = 0.16, *p* = 0.01). This study may broaden the possibility to evaluate psychological features in this clinical population; it may contribute to tailoring psychological care and related treatments in the context of health services.

## 1. Introduction

### 1.1. Fibromyalgia

Several health pathologies are described not only with medical signs and symptoms but also with the subjective perception reported by the person with the syndrome. Uncovering the specific psychological features that compose a pathology can allow better care for patients, who could also acquire a more solid awareness of what they are experiencing. Fibromyalgia is a chronic pain condition characterized by persistent and widespread musculoskeletal pain that lasts beyond three months [1]. It affects 5.0% of the world population and the incidence is greater in Europe (2.64%) compared to the US (2.41%) and Asia (1.62%), [2]. Females exhibit a markedly three times higher incidence of fibromyalgia compared to males [3].

The precise etiology of fibromyalgia remains elusive. However, the scientific literature strongly supports that it could result from the connection between fibromyalgia and central sensitization, a phenomenon linked to pain amplification. Central sensitization is characterized by an increased sensitivity to pain stimuli (hyperalgesia) and the perception of pain from typically non-painful stimuli (allodynia). This sensitivity is believed to result from dysfunction in pain modulation mechanisms, where the normal inhibitory processes that dampen pain signals are impaired [4].

Some studies also suggest a significant hereditary component in the development of the disorder and research examining the genetic profiles of families with multiple fibromyalgia cases has identified a marked genetic association, particularly on the chromosome 17p11.2-q11.2. This chromosomal region contains two genes, TRPV2 and SLC6A4, which are considered potential contributors to the syndrome’s manifestation [5]. Moreover, recent studies highlighted the cumulative effect of chronic stress as a possible significant contributor to the onset of this condition [6].

Fibromyalgia’s diagnostic process is complex mainly due to its symptomatic overlap with various other conditions. These may include inflammatory disorders like familial mediterranean fever, neurological pain syndromes, and other chronic pain conditions [7,8]. Furthermore, similar comorbidities could complicate the identification and the differentiation of fibromyalgia from related disorders. Altıntas and Melikoglu [9] demonstrated a comorbidity between familial mediterranean fever and fibromyalgia and Wang and colleagues [10] showed similar results, with other forms of physical pathologies like ulcerative colitis.

To mitigate misdiagnosis, current fibromyalgia identification protocols rely on musculoskeletal evaluation and more recent trends indicate increasingly rigorous diagnostic criteria [11]. Characteristics and criteria for the diagnosis have been outlined in previous literature, such as that by the American College of Rheumatology (ACR) (see [12] for a complete explanation).

Taken all together, when considering treatments for this clinical condition, it is crucial to choose a comprehensive vision of the disease. In this direction, applying the biopsychosocial model together with a multidisciplinary approach could help, especially because no single treatment improves functioning and minimizes the whole spectrum of fibromyalgia symptoms alone [13]. Accordingly, the biopsychosocial model highlights not only a genetic predisposition to the medical condition, but also possible alterations of emotional processing, hormonal imbalances (biological), stress, mood disorders, negative cognitive patterns (psychological), and the impact of social support, as well as cultural attitudes and economic status (social). Fibromyalgia should in fact not be understood as having a non-binary nature (normal vs. impaired), but rather by taking into account the features that characterize this “dimension” [14].

The medical treatment for fibromyalgia is frequently the preferred approach in clinical contexts, although such treatments may present several negative side effects (e.g., decreased cognitive and motor skills, dizziness, vision problems, fatigue, temporary cognitive difficulties, tremor, unusual eye spasms, dry mouth, bloating, male fertility problems or weight gain) and minor improvement when used alone [15].

Moreover, research suggests that the disability and suffering experienced by these individuals are not due to pain alone, but rather to psychological ways of responding to that pain as a threat, allowing it to become the central feature of their lives [16]. Hence, the role of an interconnected system of services for psychological treatments and healthcare is crucial for treating fibromyalgia and may help in the management of such a complex disease in combination with possible pharmacological treatments.

### 1.2. Psychological Health Services and Care for Fibromyalgia

Previous research has shown that psychotherapeutic treatments could be potentially helpful in reducing the negative symptoms of this condition and different therapeutic approaches have already been adopted in this context [17].

For instance, cognitive behavioral therapy (CBT) [18] has been used with these patients, highlighting beneficial effects especially in individuals with a juvenile onset of fibromyalgia [19]. Such results have been confirmed more recently, indicating how CBT programs may help in reducing pain catastrophizing, avoidance behaviors and hypervigilance [20,21]. In a systematic review and meta-analysis, a comparison between CBT and acceptance and commitment therapy (ACT) was made for their potential in reducing the emotional distress associated with anxiety and depression [17] and both approaches showed significant efficacy in the fibromyalgia-study participants.

In another study, focused on mindfulness-based stress reduction therapy (MBSR), 95 fibromyalgia patients were randomly assigned to a MBSR group or a waitlist group, with a related reduction in physical pain, depression and subjectively perceived stress in the participants of the MBSR group [22]. In another research project, an integrated method was applied to treat symptoms of fibromyalgia while evaluating patients’ quality of life (QoL). The results of such an approach (named “INTEGRO”) are not available yet (see [23] for a description and [24] for more details about ACT efficacy in fibromyalgia); however, this and similar integrated approaches seem highly promising.

Although each method has its merits in treating psychological symptoms associated with chronic pain syndromes, the typical psychological profile of the patient with fibromyalgia has rarely been defined in the literature. Moreover, patients normally follow several specialist visits before receiving a diagnosis and they tend to present a highly heterogeneous pattern of psychological disorders [25].

Therefore, it seems necessary to better describe the profiles of this disease, to contribute at improving and better tailoring clinical treatments. In this sense, the aim of the present research is to investigate the main psychological traits, behaviors, and symptoms expressed by these patients, as well as their typical resources and preserved functions, which in turn might lead to an improved alignment of existing therapeutic approaches.

### 1.3. Fibromyalgia Psychological Profile: State-of-the Art

One of the reasons why psychological distress associated with fibromyalgia is treated with medications is that its symptomatology may in some cases resemble that of a psychotic disorder and other clinical conditions [26], with difficulties in its diagnosis. Accordingly, organizing and implementing a non-pharmacological treatment requires a comprehensive understanding of a broader fibromyalgia clinical profile.

It has been shown in the literature that fibromyalgia patients display a defective positive affect regulation and pain acceptance, as well as a reduced ability to retain positive affect during times of intense physical pain [17,24,27]. This could be a significant obstacle in the life of these individuals, since research has shown that an increased disposition to experience compassion may favor compensatory mechanisms leading to reduced anxiety and depression [28]. Moreover, in a recent observational study, high correlations have been reported between the history of depressive symptoms and poor QoL in fibromyalgia patients [29] and these individuals have also been shown to frequently express feelings of being dismissed, overlooked, or not taken seriously [30].

Other studies have reported high levels of perfectionism in fibromyalgia. For example, in a study conducted by Ecija and colleagues [31] on 228 women with this condition, pain avoidance and task persistence were present in fibromyalgia patients with a higher tendency to perfectionism.

As one of the most pronounced symptoms in fibromyalgia is physical pain, this could also be related with the tendency of these patients to experience touch avoidance (see [32] for further description). A large number of individuals with fibromyalgia are also diagnosed with major depression disorder [33], but such a relationship is not clear yet [34]. A relationship between fibromyalgia and psychosis and chronic anxiety states has also emerged in some recent studies [35,36,37].

How probable is the risk for developing psychosis or psychotic traits in this population is unclear. However, it has been shown in a study conducted by Carta and colleagues [38], on a comparison between patients with fibromyalgia and a control group, that 59% of patients with fibromyalgia may exhibit mania, resulting in more than a double percentage compared with the control group. This finding was confirmed by other studies [29,39], in which a significant risk for suicide and for developing bipolar disorders was observed in persons with fibromyalgia. Further investigations are still needed in this research context.

Concerning qualitative aspects in fibromyalgia patients’ life, it is well known how QoL encompasses numerous interpretations and facets: according to the World Health Organization (WHO), a person’s view of their position in life, within the context of the culture and value systems in which they live, as well as their goals, expectations, standards, and worries, constitute their quality of life.

Enhancing QoL in these patients by reducing dysfunctional symptoms of fibromyalgia is the final goal of any applicable treatments. Although psychological health services seem to be able to play a role in treating fibromyalgia, a clinical profile still needs to be defined. This study aims to contribute to the existing literature by outlining the psychological traits, behaviors, and symptoms characterizing patients with fibromyalgia compared to healthy individuals (from normative samples). Based on the aforementioned literature, a range of psychological features were selected for the research: touch avoidance, anxiety, depression, self-criticism and perfectionism, positive/negative affect, quality of life and functionality, psychosis, as well as pain acceptance/catastrophizing. Touch avoidance and risk of manifesting psychosis can be considered two novel research points in the present field and in this light, the risk of manifesting psychosis will be explored in connection with the full range of the selected psychological features.

## 2. Method

This is a cross-sectional design study in which we first describe the psychological variables associated with fibromyalgia patients who participated in this study and then analyze the comparison of these variables between patients and samples of healthy individuals (normative sample). Finally, we assess the relationship between these variables and the risk of developing psychotic disorders in fibromyalgia patients, taking into consideration the Psychological Health Services.

### 2.1. Participants

This study was conducted in the context of psychological health care services. Participants were recruited at the Centre of Psychology and Functional Psychotherapy in collaboration with the University of Padua, the National Research Council (CNR) of Genoa, and the United Fibromyalgia Committee (CFU), an Italian association whose primary objective is to provide fibromyalgia sufferers with volunteer initiatives, activities and services. The CFU offered the sample of people with fibromyalgia necessary to carry out the research. All candidates involved in the study were diagnosed with fibromyalgia syndrome with a duration of the disease averaging 4.07 ± 4.21 years (range: 1–20 years) from onset to data collection in the present study.

The final sample included 76 Italian women (age range 23–78 years, mean = 50.22 ± 10.47 years). Participants with a diagnosis of psychiatric disease were not included in this study. They were asked whether they were following any possible medical treatment related with the fibromyalgia. Indeed, in this context, types of pharmacological treatments and posology are heterogeneous: none of the participants presented any critical side effects considered to directly affect the investigation. A medical physician with rheumatology specialization certified the fibromyalgia diagnosis of the patients included in the study. The participants were selected with the exclusion of psychiatric illnesses, absence of psychosis, with only women included, to investigate a part of the population with high incidence of fibromyalgia. The patients spoke Italian; foreign participants were not included in order to guarantee the possibility for the volunteers to be able participate in subsequent phases of the study with psychotherapeutic treatment in the Italian language. No participant was excluded based on the assumption of medicines. More information about the sample can be found in Appendix A.

All participants signed a formal informed consent to participate in the study. The study was conducted following the Declaration of Helsinki, and it was approved by the Ethical Committee of the Faculty of Psychology of the University of Padua (protocol No. 4044).

### 2.2. Materials and Procedure

Participants took part in the current research between February and April 2021. Following the studies mentioned in the introduction, it was decided to evaluate eight different psychological traits, behaviors, and clinical symptoms: the avoidance of being physically touched by others, anxiety, depression, the possibility of having a mental state at risk of developing psychotic disorders, self-criticism, the acceptance of chronic pain, quality of life and functionality, and the predisposition of the participants to experience positive emotional states. The tools for collecting the data were all self-reported questionnaires characterized as follows:

The Touch Avoidance Questionnaire (TAQ) [40]. The TAQ is a 37-item questionnaire examining the level of touch avoidance in different contexts, such as situations involving partners, parents, siblings, friends, professional contact, and contact with strangers. Example: “I often find it unbearable to be touched by my partner”. Participants respond on a Likert scale ranging from 1 (I completely disagree) to 5 (I completely agree). The test was validated in Italian [41] but normative data for Italian women were retrieved from another study [42]. The Cronbach reliability coefficients (alphas) for the Italian validation of the TAQ is =0.84 for the subscale “Partner”, 0.89 for the subscale “Same sex”, 0.92 for the subscale “Opposite sex”, 0.88 for the subscale “Family”, and 0.59 for the subscale “Stranger”.

State-Trait Anxiety Inventory Y-2 (STAI Y-2) [43]. The STAI Y-2 examines trait anxiety through 20 items that ask subjects to describe how they usually feel. The response scale ranges from 1 (rarely) to 4 (almost always) and higher scores indicate wider anxiety levels. Examples: “I feel inadequate” or “I wish I could be as happy as others seem to be”. The STAI Y-2 was translated and validated in Italian [44] and normative data for Italian women were retrieved by Santangelo and colleagues [45]. The Cronbach reliability coefficients (alphas) for the Italian validation of the STAI Y-2 are between 0.85 and 0.90.

Beck Depression Inventory—II (BDI-II) [46]. The BDI is a 21-item self-report questionnaire that assesses symptoms of depression. Each item is scored on a scale of 0 to 3 points, with higher scores reflecting more severe symptoms. Example of an item on “pessimism”: “I am not discouraged about my future”. BDI-II was translated and validated in Italian; it provides normative data for Italian women [47]. The Cronbach reliability coefficient (alpha) for the Italian validation of the BDI-II is =0.92.

Sixteen-item Prodromal Questionnaire (iPQ-16) (normative data in [48]). The iPQ-16 examines risk of developing a psychotic disorder within six months to up to three years. The test consists of 16 items, to which the person must answer “true” or “false” and, if true, specify the level of discomfort felt on a scale from 0 (none) to 3 (high). Example: “I have sometimes been undecided whether some things I have experienced were real or imaginary”. The test has been translated and validated in Italian [49] for a population of help-seeking Italian young adults. The Cronbach reliability coefficient (alpha) for the Italian validation of the iPQ-16 is =0.81.

Forms of Self-Criticizing/Attacking and Self-Reassuring Scale (FSCRS) [50]. The FCSRS is a 22-item questionnaire that measures self-criticism and the ability to reassure oneself when things go wrong. Participants are asked to respond on a Likert scale from 0 to 4. This measure has 3 subscales: Hated-Self (e.g., “I feel a sense of disgust with myself”), Inadequate-Self (e.g., “I feel disappointed with myself easily”), and Reassured-Self (e.g., “I can remind myself of my qualities”). The scale used [51] provides normative data for the general Italian population. The Cronbach reliability coefficient (alpha) for the Italian validation of the FSCRS is =0.84 for the sub-scale “Hated-Self”, 0.86 for the subscale “Inadequate-Self”, and 0.90 for the subscale “Reassured-Self”.

Chronic Pain Acceptance Questionnaire (CPAQ) [52]. The CPAQ is a questionnaire that estimates pain acceptance through engagement in daily activities while feeling pain and compliance to the experience of pain. The questionnaire consists of 20 items with a Likert response scale ranging from 0 to 6. Higher scores indicate a greater ability to accept pain. The Italian version of the questionnaire is validated on a population of patients experiencing chronic pain [53]. The Cronbach reliability coefficients (alphas) for the Italian validation of the CPAQ are 0.83 for “Activity Engagement” and 0.76 for “Pain Willingness”.

Fibromyalgia Impact Questionnaire (FIQ). The FIQ is a test developed to gather information on the health status and quality of life of fibromyalgia patients. It includes 20 items that measure various aspects such as physical health, psychological distress, pain, sleep, fatigue, and well-being (normative data in [54]). Example: “How did you feel when you woke up?”. The questionnaire is translated into Italian [55]. The FIQ is mainly divided into three parts: the first part aims to understand the fibromyalgia impact on simple daily activities (e.g., doing the laundry, shopping, driving a car). The second part focuses on how many days of the week the patient felt good and how many days he/she missed work or homework due to the syndrome side effects. Finally, the third examines how fibromyalgia impacts daily life and causes symptoms like “anxiety” and “tiredness”. The Cronbach reliability coefficients (alphas) for the Italian validation of the FIQ report 0.84 for “physical functioning” and 0.89 for the full questionnaire.

Dispositional Positive Emotion Scales (DPES) (normative data in [56]). The DPES assesses predisposition to experience positive emotional states. It comprises 37 items with a Likert response scale ranging from 1 to 7. The Italian version of the questionnaire has 6 subscales: “Happiness”, represents a positive attitude towards life in general (e.g., “I am an intensely cheerful person”); “Compassion” represents the desire to take care of the wellbeing of others, especially those who are vulnerable or needy (e.g., “Taking care of others gives me a warm feeling inside”); “Amusement”, refers to the ability to experience life with humor (e.g., “The people around me make a lot of jokes”); “Love”, addresses how people conceive of closeness and authenticity (e.g., “I love many people”); “Pride”, focuses on the social image of oneself (e.g., “Many people respect me”); and “Awe”, measures the ability to connect with something greater than one’s self (e.g., “I see beauty all around me”). The questionnaire was validated in Italian [57]. The Cronbach reliability coefficients (alphas) for the Italian validation of the DPES are 0.91 for the sub-scale “Happiness”, 0.80 for the sub-scale “Pride”, 0.79 for the sub-scale “Awe”, 0.80 for the sub-scale “Love”, 0.80 for the sub-scale “Compassion”, and 0.75 for the sub-scale “Amusement”.

The administration of the tools was divided into two sets to reduce the probability of fatigue, a central feature of this syndrome. The first battery consisted of TAQ, STAI, BDI and IPQ, the second consisted of FSCRS, CPAQ, FIQ and DPES. The estimated time for each battery was between 20 and 30 min. One or two days passed between one battery and another. Furthermore, examiners explained to participants that they could make pauses or stop the compilation at any time.

We state that this study is part of a larger body of research; the data presented, with new and ad hoc analyses, were also used in the first author’s dissertation.

### 2.3. Statistical Analyses

First, we conducted descriptive analyses for all subscales and subsequently, we used one-sample *t*-tests for comparing fibromyalgia patients with available normative data (the Italian healthy population for all questionnaires except the CPAQ (Chronic Pain Acceptance Questionnaire), and FIQ (Fibromyalgia Impact Questionnaire), for which normative data was only available for a population of Italian patients suffering from chronic pain and fibromyalgia, respectively). We applied Benjamini–Hochberg’s correction for multiple comparisons and used Cohen’s d as a measure of effect size. Lastly, we explored which psychological aspects were most associated with the risk of developing a psychotic disorder (i.e., iPQ-16 scores) via multiple linear regression, using as predictors all other subscales of the dataset except the CPAQ total score (which is perfectly collinear with the CPAQ subscales). Additional correlation analyses were made for an overview about the patterns of relationships among the variables used (Appendix A). Analyses were conducted using R 4.4.1.

## 3. Results

The descriptive statistics in relation to the scores obtained by the 76 patients on the clinical scales, as well as the means reported for Italian populations in the validation studies are reported in Appendix A. It should be noted, the normative population varied across questionnaires, according to data availability. Table 1 reports the results of the one-sample *t*-tests comparing the scores of the fibromyalgia patients with the normative data. All *p*-values are corrected. Means and standard errors for patients and controls are reported in Figure 1. The correlation plot is reported in the Appendix A. With regards to the TAQ, participants showed a high level of touch avoidance with people of the same sex (*t* = 26.83, df = 75, *p* < 0.001), opposite sex (*t* = 3.67, df = 75, *p* < 0.001), partners (*t* = 14.83, df = 75, *p* < 0.001), or strangers (*t* = 5.59, df = 75, *p* < 0.001). The overall profile of touch avoidance was rather different from the general population, with significantly high touch avoidance towards friends—especially same-sex friends—and partners, with lower touch avoidance towards family and strangers.

Considering the results on the scales STAI Y-2, BDI and iPQ-16, the results showed that patients with fibromyalgia have high levels of anxiety (*t* = 11.86, df = 75, *p* < 0.001), depressive symptoms (*t* = 11.62, df = 75, *p* < 0.001), and a risk of manifesting psychotic disorders (*t* = 7.66, df = 75, *p* < 0.001). With regard to self-criticism and perfectionism as measured with the FCSRS, significant differences were shown in the domains of hated-self (*t* = 2.78, df = 75, *p* < 0.01) and reassured-self (*t* = −5.91, df = 75, *p* < 0.001), but not in the domain of inadequate-self. Patients with fibromyalgia showed a high tendency towards self-hate thoughts and a low tendency towards self-reassuring thoughts. In all cases, effect sizes were small to medium.

Pain willingness measured with the CPAQ was significantly lower for patients with fibromyalgia than for patients with chronic pain (*t* = −2.98, df = 75, *p* < 0.01). Considering the FIQ, which is specific for patients with fibromyalgia, participants did not show a significant perception of poor functionality in their everyday life, but they generally suffered from scarce quality of life (e.g., in terms of impact on physical functioning, number of days in which they felt good, and number of days of missed work) (*t* = 6.34, df = 75, *p* < 0.001). Finally, participants showed fewer positive emotions compared to controls (DPES), except for compassion (*t* = 5.94, df = 75, *p* = 0.09) which was considerably higher in patients with fibromyalgia (no significant difference shown when considering the “awe” dimension).

Regarding predictors of the scores on the iPQ-16, multiple linear regression included all other subscales of the dataset, apart from the total score on the CPAQ (which would be collinear with its two subscales). The results highlighted a few significant predictors of the iPQ-16 score. High depressive symptoms (*B* = 0.49, *t* = 0.20, *p* = 0.01), low activity engagement (*B* = 0.41, *t* = 0.13, *p* = 0.02), high touch avoidance with strangers (*B* = −0.20, *t* = 0.14, *p* = 0.01), poor physical functioning (*B* = 0.45, *t* = 0.14, *p* < 0.01), and low quality of life (*B* = −0.40, *t* = 0.16, *p* = 0.01) predicted the risk of manifesting psychotic disorders, Table 2.

## 4. Discussion

This study investigated the psychological features of individuals with fibromyalgia, considering a range of selected traits, behaviors, and clinical symptoms that were hypothesized to have a role in composing the clinical profiles of these individuals. The data collected on the psychological functioning in a group of individuals with fibromyalgia were compared with relative normative samples. Among the psychological aspects assessed, the degree of association with the risk of developing a psychotic disorder (measured with the iPQ-16 scores) was explored.

The results showed that the group of individuals with fibromyalgia had significantly higher levels of anxiety and depressive symptoms compared to healthy individuals, which is consistent with the findings of previous research [58]. This confirms that fibromyalgia is related to a wide range of psychological features, which, from a therapeutic perspective, is particularly crucial to consider when treating these patients.

Notably, the individuals with fibromyalgia in the present study were administered the Touch Avoidance Questionnaire (TAQ) [41] and results showed a significantly high touch avoidance level. A possible interpretation of this finding is that, given physical discomfort as a primary symptom of fibromyalgia, these individuals might prefer to avoid straight touch with its potential impact on social interaction and quality of life [59]. This result could also be interpreted by further considering aspects such as cultural factors (e.g., attitudes and traits typical of a culture can influence the disposition towards physical contact) and tool-specific characteristics that could have intervened. Thus, more research will be necessary in this direction.

Findings also revealed that, compared to people suffering from pathological chronic pain (non-fibromyalgic pain), participants in this study exhibited a significantly reduced ability to cope with pain, especially in their everyday activities. Moreover, individuals with fibromyalgia showed a significantly high rate of self-criticism and reduced capacity to reassure themselves, with an overall poor quality of life compared with controls, aspects that reflect the previous results well [59,60,61]; for example, Singh and collaborators, in 2024, showed a considerable level of mood disturbances, somatization, and sleep disorders in fibromyalgia patients at a severe stage of the syndrome. However, a direct relationship between such conditions and pain in these patients could not be determined; to outline and track the “behavior” of such disease, it would be necessary to include a longitudinal monitoring of participants’ health conditions.

With regards to the disposition to experience positive emotional states in different contexts, the results uncovered that patients with fibromyalgia experience a different disposition to happiness, pride, amusement, and compassion, compared to individuals who do not suffer from the same disease. Notably, while the perceived happiness, pride, and amusement was lower compared to the healthy population, the degree of compassion was higher.

These patients seem to some degree characterized by a pronounced sense of empathy (estimated in terms of their disposition to take care of the wellbeing of others, especially towards persons who are vulnerable or needy); this result also had some points in common with a previous study in which women with fibromyalgia were found to display significant empathy capacities [62]. It would be interesting to analyze how this disposition could be perceived in relation to the period before and after the onset of the disease. From these data, the results could be interpreted in relation to the possible necessity for these patients to cope with a clinical condition characterized by physical pain, which may indirectly influence the ability to empathize with others’ emotional states.

The degree of association of patients’ clinical features with the risk of developing a psychotic disorder showed that participants with higher rates of psychosis risk were more sensitive to touch (higher avoidance) and presented more severe symptoms of depression, accompanied by poor involvement in daily activities and overall poor quality of life. Although the relationship between fibromyalgia symptoms and possible psychotic traits has been investigated in previous studies [36,37], how to act in terms of prevention, from a psychological point of view, is still unclear. Integrating the results of this study with those previously reported in the literature, it could be suggested that people with fibromyalgia who are characterized by higher levels of depression, touch avoidance and inactivity might be significantly exposed to the risk of manifesting psychotic suffering.

Although we are aware that these results deserve further confirmation, replication and robustness in the methodology, we hope this study sheds light on some of the psychological sufferings that could be part of fibromyalgia. Depressive symptoms, accompanied by traits of avoidance and poor involvement in daily activities might have an “interactive role” with some precursors of psychosis. More research is needed to understand how these aspects may vary depending on a gradient of severity of fibromyalgia symptoms.

Overall, the results of this study highlight (1) the necessity to assess the psychological profiles of people with fibromyalgia through a multi-componential perspective and (2) consider in preventive terms, the potential risk of psychosis in relationship with depression, touch avoidance and symptoms like apathy or scarce motivation. The purpose is to contribute, by better understanding this syndrome, to possibly implement tailored treatments for these patients in the context of clinical health services.

## 5. Limitations

Although self-report is preferred when the main interest is associated with a subjective perception of disability/dysfunction, this method can sometimes lead patients (especially those suffering from physical pain) to overestimate the severity of their condition or under-report it. Overall, the results pertain to the questionnaires used and some degrees of limitation can be expected in their ability to fully capture the psychological complexities of fibromyalgia patients. Moreover, the research focuses on the psychological features of fibromyalgia in Italian women and thus, the results should not be considered as exhaustive for other sex orientations or for those who live in other cultural contexts; cross-cultural diversity should be considered in this sense. This study illustrates psychological features, and future research could integrate them with physiological aspects related with the disease (e.g., parameters related with sleep quality or other measures). We also believe that future research could address more homogeneous data concerning the disease duration and a control group more directly related to the experimental group, as well as an evaluation about psychological functioning on a larger sample, comparing different cultural backgrounds, to increase knowledge on this topic and enlarge the robustness of the results.

## 6. Conclusions

Patients with fibromyalgia exhibited psychological dysfunctions in physical touch by others, anxiety and depression; risk of developing psychotic disorders, self-criticism and perfectionism, acceptance of chronic pain, impact on quality of life and predisposition to experience positive emotional states. In these patients, a higher risk of experiencing psychosis was related to more severe symptoms of depression, poor involvement in daily activities, touch avoidance and overall poor quality of life. These data show that treating symptoms of fibromyalgia sufferers through psychological interventions may require a broad view on the co-occurrent clinical conditions, to target psychological interventions. Further studies may address socio-demographic, emotional and relational capacities in these individuals, as such variables may play a role in modulating such clinical conditions. Lastly, addressing possible risk factors in this context could be helpful for prevention purposes.

## Figures and Tables

**Figure 1 behavsci-14-01016-f001:**
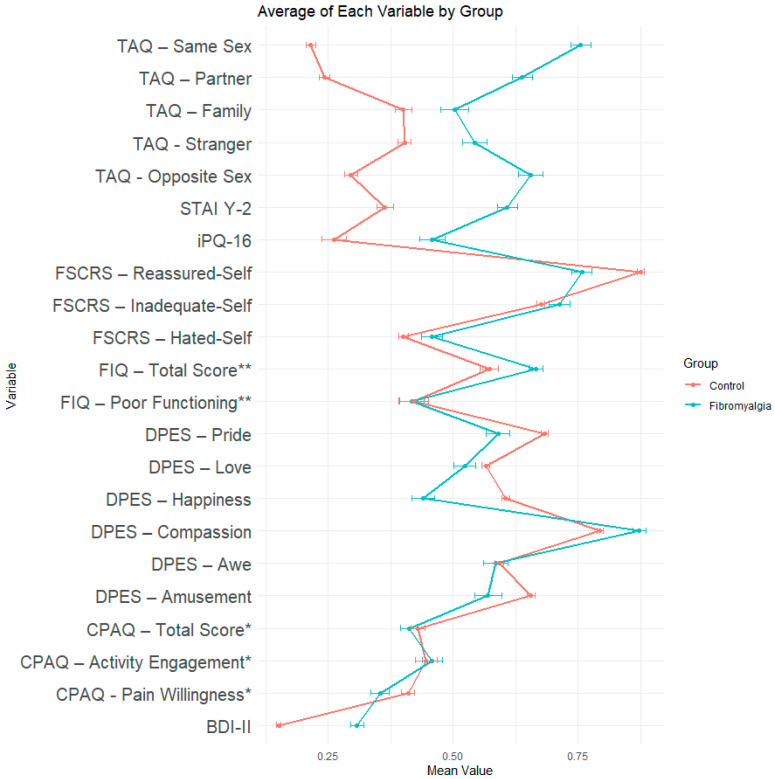
Means and standard errors for patients with fibromyalgia and healthy population. * = control scores on the CPAQ refer to a population affected by chronic pain; ** = Control scores on the FIQ refer to another sample of patients with fibromyalgia. All scale scores are rescaled to have a theoretical minimum score of 0 and a theoretical maximum score of 1. In TAQ, STAI Y-2, iPQ-16, FIQ, and BDI-II: higher scores in the table reflect worse disposition/condition in patients compared to controls; in the FSCRS Reassured-Self, also in all the DPES and CPAQ scales: higher scores indicate better functioning/psychological disposition.

**Table 1 behavsci-14-01016-t001:** One Sample *t*-Test of the scales used to assess the patients with fibromyalgia. *t*-test, degrees of freedom, *p*-value, and Cohen’s d are reported in the table.

Scale/Subscale	*t*-Test	df	*p*-Value	Cohen’s *d*
**TAQ**				
*TAQ—Partner*	19.22	74	<0.001	2.22 (large)
*TAQ—Family*	3.67	75	<0.001	0.42 (small)
*TAQ—Same Sex*	26.83	75	<0.001	3.08 (large)
*TAQ—Opposite Sex*	14.83	75	<0.001	1.70 (large)
*TAQ—Stranger*	5.59	75	<0.001	0.64 (medium)
**STAI Y-2**	11.86	75	<0.001	1.36 (large)
**BDI-II**	11.62	75	<0.001	1.33 (large)
**iPQ-16**	7.66	75	<0.001	0.88 (large)
**FSCRS**				
*FSCRS—Hated-Self*	2.78	75	0.009	0.32 (small)
*FSCRS—Inadequate-Self*	1.7	75	0.115	0.19 (small)
*FSCRS—Reassured-Self*	−5.91	75	<0.001	0.68 (medium)
**CPAQ**				
*CPAQ—Pain Willingness*	−2.98	75	0.006	0.34 (small)
*CPAQ—Activity Engagement*	0.63	75	0.584	0.07 (negligible)
*CPAQ—Total Score*	−1.09	75	0.324	0.12 (negligible)
**FIQ**				
*FIQ—Poor Functioning*	−0.12	75	0.906	0.01 (negligible)
*FIQ—Total Score*	6.34	75	<0.001	0.73 (medium)
**DPES**				
*DPES—Happiness*	−7.04	75	<0.001	0.81 (large)
*DPES—Pride*	−3.95	75	<0.001	0.45 (small)
*DPES—Love*	−1.83	75	0.092	0.21 (small)
*DPES—Compassion*	5.94	75	<0.001	0.68 (medium)
*DPES—Amusement*	−3.19	75	0.003	0.37 (small)
*DPES—Awe*	−0.25	75	0.844	0.03 (negligible)

**Table 2 behavsci-14-01016-t002:** Predictors of the scores on the iPQ-16. The results of the multiple linear regression in the table show the (non-causal) relationship between clinical scores (independent variables, first column) and risk of manifesting psychotic disorders (dependent variable).

Scale/Subscale	Std. β	*t*	*p*-Value
**TAQ**			
*TAQ—Partner*	−0.02	0.1	0.966
*TAQ—Family*	0.04	0.1	0.813
*TAQ—Same Sex*	−0.1	0.12	0.699
*TAQ—Opposite Sex*	0.31	0.12	0.419
*TAQ—Stranger*	−0.2	0.14	0.013
**STAI Y-2**	−0.03	0.2	0.888
**BDI-II**	0.49	0.2	0.015
**FSCRS**			
*FSCRS—Hated-Self*	0.03	0.15	0.839
*FSCRS—Inadequate-Self*	0.34	0.17	0.049
*FSCRS—Reassured-Self*	−0.28	0.16	0.09
**CPAQ**			
*CPAQ—Pain Willingness*	−0.15	0.11	0.192
*CPAQ—Activity Engagement*	0.41	0.13	0.002
**FIQ**			
*FIQ—Poor Functioning*	0.45	0.14	0.002
*FIQ—Total Score*	−0.4	0.16	0.013
**DPES**			
*DPES—Happiness*	−0.1	0.14	0.475
*DPES—Pride*	−0.05	0.16	0.747
*DPES—Love*	−0.07	0.13	0.55
*DPES—Compassion*	0.09	0.11	0.388
*DPES—Amusement*	0.05	0.11	0.647
*DPES—Awe*	0.12	0.12	0.318

## Data Availability

Data of this study are not publicly available due to privacy reasons.

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
