# Peer review of "Psychological Features of Fibromyalgia in the Psychological Health Services"

_behavsci, 2024, doi:10.3390/bs14111016_

Round 1
Reviewer 1 Report
Comments and Suggestions for Authors
This is a relevant topic as it is important to strengthen the psychological aspect of the biopsychosocial model in fibromyalgia care for a comprehensive intervention.
This paper is part of the main author's Ph.D. thesis and therefore has a high degree of plagiarism.
The title indicates the content of the study in a clear and direct way.
The abstract allows a quick identification of the basic content, although it does not present results with numerical values.
The literature review is outdated, only 36% of the referenced articles are from the last five years.
In terms of structure, there are too many tables in the results and the conclusion section is missing.
The theoretical framework in the introduction does not mention the biopsychosocial model.
The methodology does not describe the design chosen. The data collection procedure is not described. The reliability and validity of the instruments used are not described.
The results are presented only in tables and figures. The tables are very long and go beyond the space of the text. The legend explaining the tables and figures is too long.
The interpretation of the discussion is based on the data. The limitations of the study are not discussed. The conclusions section is missing in relation to the objectives of the study.
Referencias son adecuados, aunque solo el 36% son de hace cinco años. El año se da en al final de la referencia (véase la referencia 30). Citas propias en las referencias 29 y 30.
Author Response
Comments - Reviewer 1
This is a relevant topic as it is important to strengthen the psychological aspect of the biopsychosocial model in fibromyalgia care for a comprehensive intervention.
REPLY: the importance of considering the biopsychosocial model is largely integrated now as suggested by the reviewer (page 3, lines 30 to 40).
2) This paper is part of the main author's Ph.D. thesis and therefore has a high degree of plagiarism.
REPLY: Although the main author wrote, indeed, her thesis on this dataset, the present paper revised both the text and the analyses; moreover, the new research investigation is not previously published (pages 3-15). We have also reported it in the method section of the paper (pages 8, 35-36).
3) The title indicates the content of the study in a clear and direct way.
REPLY: the title is now changed. We hope to interpret correctly the reviewer’s suggestion.
4) The abstract allows a quick identification of the basic content, although it does not present results with numerical values.
REPLY: as abstracts are in most of the cases preferred without numerical values, we did not report them in the initial version. Moreover, the abstracts of the papers previously published in this special issue of the Journal did not report numerical values. However, to encounter the request of the reviewer we added numerical values where possible (page 2, lines 16 to 20).
5) The literature review is outdated, only 36% of the referenced articles are from the last five years.
REPLY: We further added other updated papers, which have been now added and integrated in this new version (page 3-6).
6) In terms of structure, there are too many tables in the results and the conclusion section is missing.
REPLY: Thanks for the suggestion. To lighten the result section, the descriptive table is now moved to the supplementary materials (page 10-13 and supplementary materials table S2 which was previously Table 1).
The conclusion section is reported now, in a proper section. (page 15)
7) The theoretical framework in the introduction does not mention the biopsychosocial model.
REPLY: Thank you for this comment. We now have integrated this important aspect with the text (page 3, lines 30 to 40)
8) The methodology does not describe the design chosen. The data collection procedure is not described. The reliability and validity of the instruments used are not described.
REPLY: Thank you for this comment. We now added a short introduction about the design used in the study (page 6, lines 10 - 15). For the details about the reliability and validity of each instrument we need to rely on the original papers, however we reported in the manuscript now for each of the instruments the associated Cronbach value (pages 7 and 8).
9) The results are presented only in tables and figures. The tables are very long and go beyond the space of the text. The legend explaining the tables and figures is too long.
REPLY: Thank you for this comment. The results are now reported in the text in numerical form. To note, in this study we followed the "rule of thumb" of making figures and tables possibly self-standing: this approach, as adopted in other papers, can be useful in terms of readability, i.e., to agilely catch the content in a table or figures from the caption. In agreement with the reviewer, we now have shortened some captions. Notably, the caption of figure 1 is necessarily longer than the others as some values were reversed and scaled to make it visually clearer and more readable (page 12, lines 1 to 7).
10) The interpretation of the discussion is based on the data. The limitations of the study are not discussed. The conclusions section is missing in relation to the objectives of the study.
REPLY: The Discussion is now more extensively elaborated to provide a deeper analysis of the results and their implications, addressing potential limitations and suggesting areas for future research. We now integrated the paper with a separate Limitation section. We now integrated the paper with a Conclusion section (pages 13-15).
11) Referencias son adecuados, aunque solo el 36% son de hace cinco años. El año se da en al final de la referencia (véase la referencia 30). Citas propias en las referencias 29 y 30.
REPLY: We struggled to understand this comment which is Spanish. Most of our references are dated between 2018-2024. We now modified reference n.30 and added some other recent references to the paper. Furthermore, references (29 and 30) were strictly necessary, since authors Casetta, Rizzi and Passarelli translated and validated the Touch Avoidance Questionnaire to Italian.

Reviewer 2 Report
Comments and Suggestions for Authors
Dear Editor
As a scientist I, agree 100 % with the results and importance of the topic in the manuscript,
The manuscript presents a significant study on the prevalence of fibromyalgia among Italian females, focusing on the relationship between constant pain results firm fibromyalgia and behavioral reflections such as pain , depression, etc. However, the field of fibromyalgia research includes numerous factors that should be addressed.
1-The introduction should include a discussion on the overlapping symptoms of fibromyalgia and Familial Mediterranean Fever (FMF), particularly in Mediterranean populations such as Italians. It is crucial to mention the genetic predispositions, such as inherited family medical conditions like estrogen receptor-positive breast cancer in females( the author should take family medical history for every single paitent).
2- the exclusion process for other overlapping and underlying illnesses and underlying disorders that accompany fibromyalgia, such as Ulcerative Colitis and neurological pain, should be detailed.
3-The author should also pay close attention to the introduction and discussion sections, ensuring they comprehensively cover these aspects.
4-Furthermore, referencing other studies that measure laboratory parameters in fibromyalgia patients will enhance the value of the work such as (A biochemical study of chronic stress and chronic inflammation fibromyalgia. Pharm Pharmacol Int J. 2018;6(3):234‒243. DOI: 10.15406/ppij.2018.06.00181 ) is very helpful manuscript.
This manuscript link between psychological and physiological aspects of fibromyalgia
5- the author should mention the study limitations such as:
- The specific questionnaires used may have limitations in their ability to fully capture the psychological complexities of fibromyalgia patients
-Population number, sex, race, etc,,,
- the study provides illustrated into psychological features, it may not fully address the interplay between psychological and physiological aspects of fibromyalgia
Comments on the Quality of English LanguageDear Editor
As a scientist I, agree 100 % with the results and importance of the topic in the manuscript,
The manuscript presents a significant study on the prevalence of fibromyalgia among Italian females, focusing on the relationship between constant pain results firm fibromyalgia and behavioral reflections such as pain , depression, etc. However, the field of fibromyalgia research includes numerous factors that should be addressed.
1-The introduction should include a discussion on the overlapping symptoms of fibromyalgia and Familial Mediterranean Fever (FMF), particularly in Mediterranean populations such as Italians. It is crucial to mention the genetic predispositions, such as inherited family medical conditions like estrogen receptor-positive breast cancer in females( the author should take family medical history for every single paitent).
2- the exclusion process for other overlapping and underlying illnesses and underlying disorders that accompany fibromyalgia, such as Ulcerative Colitis and neurological pain, should be detailed.
3-The author should also pay close attention to the introduction and discussion sections, ensuring they comprehensively cover these aspects.
4-Furthermore, referencing other studies that measure laboratory parameters in fibromyalgia patients will enhance the value of the work such as (A biochemical study of chronic stress and chronic inflammation fibromyalgia. Pharm Pharmacol Int J. 2018;6(3):234‒243. DOI: 10.15406/ppij.2018.06.00181 ) is very helpful manuscript.
This manuscript link between psychological and physiological aspects of fibromyalgia
5- the author should mention the study limitations such as:
- The specific questionnaires used may have limitations in their ability to fully capture the psychological complexities of fibromyalgia patients
-Population number, sex, race, etc,,,
- the study provides illustrated into psychological features, it may not fully address the interplay between psychological and physiological aspects of fibromyalgia
Author Response
Comments – Reviewer 2
Dear Editor
As a scientist I, agree 100 % with the results and importance of the topic in the manuscript,
The manuscript presents a significant study on the prevalence of fibromyalgia among Italian females, focusing on the relationship between constant pain results firm fibromyalgia and behavioral reflections such as pain, depression, etc. However, the field of fibromyalgia research includes numerous factors that should be addressed.
- The introduction should include a discussion on the overlapping symptoms of fibromyalgia and Familial Mediterranean Fever (FMF), particularly in Mediterranean populations such as Italians. It is crucial to mention the genetic predispositions, such as inherited family medical conditions like estrogen receptor-positive breast cancer in females (the author should take family medical history for every single patient).
REPLY: WE thank the reviewer for this comment. The important role of genetic predispositions and similar conditions to fibromyalgia with overlapping symptoms are described further in the introduction (page 3, lines 12-19).
- the exclusion process for other overlapping and underlying illnesses and underlying disorders that accompany fibromyalgia, such as Ulcerative Colitis and neurological pain, should be detailed.
REPLY: such data has been added to the article, thank you (page 3, lines 20-29).
- The author should also pay close attention to the introduction and discussion sections, ensuring they comprehensively cover these aspects.
REPLY: both the introduction and the discussion sections have been integrated with further and recent literature studies.
- Furthermore, referencing other studies that measure laboratory parameters in fibromyalgia patients will enhance the value of the work such as (A biochemical study of chronic stress and chronic inflammation fibromyalgia. Pharm Pharmacol Int J. 2018;6(3):234‒243. DOI: 10.15406/ppij.2018.06.00181) is a very helpful manuscript.
REPLY: this and other studies have been reported now in the manuscript (reference N 5 presented both in the text and in the reference list), thank you.
This manuscript link between psychological and physiological aspects of fibromyalgia
5) the author should mention the study limitations such as:
- The specific questionnaires used may have limitations in their ability to fully capture the psychological complexities of fibromyalgia patients
-Population number, sex, race, etc,,,
- the study provides illustrated into psychological features, it may not fully address the interplay between psychological and physiological aspects of fibromyalgia
REPLY: the insights proposed have been taken into consideration and integrated with the rest of the article now, thank you (page 14-15).

Reviewer 3 Report
Comments and Suggestions for Authors
This article is acceptable, although it has some important biases or areas for improvement:
METHODOLOGY:
- a very important bias is the possible influence of medication on health status, they do not record the medication that women with fibromyalgia take and side effects on mood, sleep and rest or directly on the object of study.
- Another important aspect that I do not see recorded is how long they have been diagnosed with fibromyalgia and if this has any influence on the objective of the study, a woman with 1 year of diagnosis is not the same as a woman who has been diagnosed for 5 years.
LIMITATIONS: it is strange that the large number of questionnaires used is not recorded, as is the time they have spent carrying them out, since the study is cross-sectional, using so many and so much time can cause excessive fatigue in the study participants.
Author Response
REVIEWER 3
METHODOLOGY:
- a very important bias is the possible influence of medication on health status, they do not record the medication that women with fibromyalgia take and side effects on mood, sleep and rest or directly on the object of study.
Reply: We carefully analysed the information useful for conducting the study: we asked not only if participants took medicine for fibromyalgia but also which medicines (so we considered all the types of medicines that people used at that time). We added some considerations in the method section of this article, on the potential consequences of the consumption of these medicines and how they can interfere with the results presented (page 7, lines 1-5). No participant was excluded based on the assumption of medicines. More information can be found on the first table of the supplementary material (Table S1) and we have included a reference to the supplementary material in the manuscript itself.
- Another important aspect that I do not see recorded is how long they have been diagnosed with fibromyalgia and if this has any influence on the objective of the study, a woman with 1 year of diagnosis is not the same as a woman who has been diagnosed for 5 years.
Reply: We thank the reviewer and have included this data in the participants section, in the method, page 6, lines 41-44). Moreover, we added the limitation of a large range of disease duration in its proper section in the manuscript (page 16, lines 40-41). This information can be found in Table S1 of the supplementary material with its reference in the manuscript itself.
LIMITATIONS: It is strange that the large number of questionnaires used is not recorded, as is the time they have spent carrying them out, since the study is cross-sectional, using so many and so much time can cause excessive fatigue in the study participants.
Reply: The questionnaires were administered two times and randomized to reduce the effect of the length of the questionnaires. In addition, materials were divided to reduce the probability of fatigue, a characteristic of this syndrome. This aspect was largely occupied in the administration; indeed, these are patients who have been taken care of from a psychological point of view and we have included a section in the manuscript on this aspect (page 9, lines 28-33), thank you.

Reviewer 4 Report
Comments and Suggestions for Authors
Firstly, in the summary at L19 P1, it is mentioned that fibromyalgia is a disorder that presents with memory problems. This is a very reductionist version of the cognitive issues faced by these patients (which are considered by patients as one of the central symptoms of fibromyalgia and more disabling than the pain itself), as these issues include both lower-order cognitive problems (attention, memory, and information processing speed) and higher-order cognitive problems (executive functions). Therefore, it is suggested to replace the term 'memory' with 'cognitive deficits.
Secondly, the keywords in the summary could include at least five, reflecting the fundamental variables of the work. In this sense, the word 'evaluation' is included, but this is extremely unspecific: evaluation of what type? (for example: neuropsychological evaluation? psychological evaluation?). Please specify.
Thirdly, in the introduction section, during the second paragraph, when fibromyalgia is described, it is mentioned in L42 that it is a syndrome, but this had not been previously stated. Please indicate this from the outset. See the studies by the American College of Rheumatology (ACR) on this matter. These studies are essential. Example:
Wolfe, F., Clauw, D.J., Fitzcharles, M.A., Goldenberg, D.L., Katz, R.S., Mease, P., Russell, A.S., Russell, I.J., Winfield, J.B., & Yunus, M.B. (2010). Preliminary diagnostic criteria for fibromyalgia and measurement of symptom severity by the American College of Rheumatology. Arthritis Care & Research, 62(5), 600–610. https://doi.org/10.1002/acr.20140
Likewise, when describing fibromyalgia, it would be highly recommended to quantify the global prevalence of the disease.
Continuing in the introduction, regarding etiology, although the current etiology and pathophysiology of fibromyalgia are unknown, the relationship between fibromyalgia and the phenomenon of central sensitization to pain (related to allodynia and hyperalgesia) and with aberrant circuits in pain inhibition mechanisms is well-documented in scientific literature. This is essential in the description of the disease's etiology. Please include a brief description of it.
Lastly, regarding the introduction, in L2 P3, when referring to the study that conducted a systematic review and meta-analysis comparing CBT with ACT and the reduction of emotional distress (Cojocaru et al., 2024), for which group and/or clinical condition was this true? For the general population, fibromyalgia, chronic pain...? Please specify. Additionally, when exploring the psychological profile in the state of the art, the possible relationship between fibromyalgia and psychosis is addressed too generically and loosely. In my opinion, the possible connection between fibromyalgia and the experience of psychosis is not clearly explained.
On the other hand, in the Methodology section, specifically in the Participants section, when referring to the Fibromyalgia Unified Committee (CFU), could you briefly explain what it is? Is it a public institution dependent on the state? Please specify. Additionally, continuing with the Participants section, it is noted that 76 women diagnosed with fibromyalgia at some point in the past participated. I have several questions related to this: first, who was responsible for diagnosing fibromyalgia in these patients? Were the diagnostic criteria for fibromyalgia established by the ACR followed? When in the past were they diagnosed? Please clarify. Furthermore, nothing is specified about the use of a control group (as can later be observed in the results) or the clinical conditions selected for it. This omission causes confusion in the reading of the article, making it necessary to provide a clarifying analysis. Lastly, related to the Participants section, the exclusion criteria should be specified.
As for the Materials section of the Method, regarding the description of each of the measurement instruments provided, it is necessary to specify the documentary sources that contain the psychometric data, such as Cronbach’s alpha, for each of them. In this section, considering that clinical pain is widely known as a central symptom of fibromyalgia, it would have been very interesting to assess it using specific measurement instruments like the McGill Pain Questionnaire. This questionnaire thoroughly explores the pain experienced by chronic pain patients. Another question related to the assessment of these patients' quality of life: which questionnaire was used for this evaluation?
Regarding the Statistical Analysis section of the Method, a more detailed description is necessary. For example, with the t-tests, were t-tests conducted for related or unrelated samples? Additionally, if multiple comparisons were made using the Benjamini-Hochberg correction, this should be clearly specified. It should also be stated that effect sizes were evaluated using Cohen's d. The same applies to the description of the regression analysis: it is necessary to specify the parameter used for the prediction of one variable over another. Lastly, regarding the regression analysis again, how many predictor variables were introduced? On another note, but still related to the statistical analysis, it is unclear why more robust statistical analyses were not conducted (e.g., MANOVAs, correlations, or even mediation analyses). Could you please shed some light on this? Additionally, it is necessary to indicate which statistical software was used and its version.
Lastly, in relation to the Method section, the absence of a brief description of the Procedure stands out (e.g., where and by whom was the evaluation conducted? How many sessions? How many hours per session? Did the sessions include breaks? What was done in each of the sessions?). Regarding what is specified in the Discussion, I believe that the relationship between the clinical characteristics of the patients and the possible occurrence of psychosis needs a more in-depth qualitative analysis, as this was introduced by the authors as a novel aspect of the study; however, the information derived from the analyses conducted is too limited. The relationship between them is not clearly explained. In conclusion, it would be interesting for the authors to further explore how the observed findings could directly influence the psychological treatment of the participants.
Author Response
REVIEWER 4
- Firstly, in the summary at L19 P1, it is mentioned that fibromyalgia is a disorder that presents with memory problems. This is a very reductionist version of the cognitive issues faced by these patients (which are considered by patients as one of the central symptoms of fibromyalgia and more disabling than the pain itself), as these issues include both lower-order cognitive problems (attention, memory, and information processing speed) and higher-order cognitive problems (executive functions). Therefore, it is suggested to replace the term 'memory' with 'cognitive deficits.
Reply: Fibromyalgia is characterized by a wider range of cognitive symptoms and there is even a term for this feature (i.e., "fibro-fog"). We thank the reviewer for this correction and we have corrected it in the manuscript, as suggested.
- Secondly, the keywords in the summary could include at least five, reflecting the fundamental variables of the work. In this sense, the word 'evaluation' is included, but this is extremely unspecific: evaluation of what type? (for example: neuropsychological evaluation? psychological evaluation?). Please specify.
Reply: we have detected three keywords considered in line with the study and they have been indicated in the title page, thank you (we followed journal guidelines recommending three to ten keywords, thank you).
- Thirdly, in the introduction section, during the second paragraph, when fibromyalgia is described, it is mentioned in L42 that it is a syndrome, but this had not been previously stated. Please indicate this from the outset. See the studies by the American College of Rheumatology (ACR) on this matter. These studies are essential. Example: Wolfe, F., Clauw, D.J., Fitzcharles, M.A., Goldenberg, D.L., Katz, R.S., Mease, P., Russell, A.S., Russell, I.J., Winfield, J.B., & Yunus, M.B. (2010). Preliminary diagnostic criteria for fibromyalgia and measurement of symptom severity by the American College of Rheumatology. Arthritis Care & Research, 62(5), 600–610. https://doi.org/10.1002/acr.20140 Likewise, when describing fibromyalgia, it would be highly recommended to quantify the global prevalence of the disease.
Reply: We have inserted references to these important aspects (page 3, lines 7-11), thank you for this contribution.
- Continuing in the introduction, regarding etiology, although the current etiology and pathophysiology of fibromyalgia are unknown, the relationship between fibromyalgia and the phenomenon of central sensitization to pain (related to allodynia and hyperalgesia) and with aberrant circuits in pain inhibition mechanisms is well-documented in scientific literature. This is essential in the description of the disease's etiology. Please include a brief description of it.
Reply: We thank the reviewer for this clarification and we think that he has given an added value to this part, inserted in the introduction on page 3 (lines 12-18).
- Lastly, regarding the introduction, in L2 P3, when referring to the study that conducted a systematic review and meta-analysis comparing CBT with ACT and the reduction of emotional distress (Cojocaru et al., 2024), for which group and/or clinical condition was this true? For the general population, fibromyalgia, chronic pain...? Please specify.
Reply: as specified in the manuscript (page 4, line 38), this was true for participants with fibromyalgia referred in the studies analyzed in the meta-analysis. Thank you.
- Additionally, when exploring the psychological profile in the state of the art, the possible relationship between fibromyalgia and psychosis is addressed too generically and loosely. In my opinion, the possible connection between fibromyalgia and the experience of psychosis is not clearly explained.
Reply: A better description of this aspect, supported by results present in the literature has been reported in the paper (page 6-7; lines 41-44, 1-3), thank you.
- On the other hand, in the Methodology section, specifically in the Participants section, when referring to the Fibromyalgia Unified Committee (CFU), could you briefly explain what it is? Is it a public institution dependent on the state? Please specify.
Reply: We thank the reviewer; this aspect was reported in the text (page 6, lines 36-40): where we report that participants were recruited at the Center of Psychology and Functional Psychotherapy in collaboration with the University of Padua, the National Research Council (CNR) of Genoa, and the United Fibromyalgia Committee (CFU), an Italian Association whose primary objective is to provide fibromyalgia sufferers with volunteer initiatives, activities and services. CFU offered the sample of people with fibromyalgia necessary to carry out the research. https://www.cfuitalia.it/
Additionally, continuing with the Participants section, it is noted that 76 women diagnosed with fibromyalgia at some point in the past participated. I have several questions related to this: first, who was responsible for diagnosing fibromyalgia in these patients? Were the diagnostic criteria for fibromyalgia established by the ACR followed? When in the past were they diagnosed? Please clarify. Furthermore, nothing is specified about the use of a control group (as can later be observed in the results) or the clinical conditions selected for it. This omission causes confusion in the reading of the article, making it necessary to provide a clarifying analysis. Lastly, related to the Participants section, the exclusion criteria should be specified.
Reply:
- For this research, we adopted the diagnosis criteria provided by the Certification of Disease by a Rheumatologist (a specialized doctor recognized in the Italian healthcare context). In general, according to the Italian healthcare system, it is the medical doctor who certifies the diagnosis to the patient with fibromyalgia. The diagnosis is certified by the rheumatologist.
- The control group is the normative sample of the instrument used (abstract, introduction, method, statistical analyses, results and the discussion clarify make this aspect transparent). To this revised version, we also included this aspect as a limit of the study (with no direct control group).
- The participants of this study were selected with the exclusion of psychiatric illnesses and psychosis, only women were enrolled; all patients spoke Italian (page 7, lines 12-18)
- As for the Materials section of the Method, regarding the description of each of the measurement instruments provided, it is necessary to specify the documentary sources that contain the psychometric data, such as Cronbach’s alpha, for each of them.
In this section, considering that clinical pain is widely known as a central symptom of fibromyalgia, it would have been very interesting to assess it using specific measurement instruments like the McGill Pain Questionnaire. This questionnaire thoroughly explores the pain experienced by chronic pain patients. Another question related to the assessment of these patients' quality of life: which questionnaire was used for this evaluation?
Reply: the article shows the Cronbach and related references for all the tools at pages 7-8-9, that is in the section in which materials are described.
The McGill Pain Questionnaire will be one of the tools we could use in future studies, and we thank the reviewer for the suggestion.
Quality of life was measured using the Fibromyalgia Impact Questionnaire. The FIQ is mainly divided into three parts: the first part aims to understand the fibromyalgia impact on simple daily activities (e.g., doing the laundry, shopping, driving a car). The second part focuses on how many days of the week the patient felt good and how many days he/she missed work or homework due to the syndrome side effects. The third part studies how fibromyalgia impacts daily life and causes certain symptoms like anxiety and tiredness (page 9, lines 6-9).
Regarding the Statistical Analysis section of the Method, a more detailed description is necessary. For example, with the t-tests, were t-tests conducted for related or unrelated samples? Additionally, if multiple comparisons were made using the Benjamini-Hochberg correction, this should be clearly specified. It should also be stated that effect sizes were evaluated using Cohen's d. The same applies to the description of the regression analysis: it is necessary to specify the parameter used for the prediction of one variable over another. Lastly, regarding the regression analysis again, how many predictor variables were introduced? On another note, but still related to the statistical analysis, it is unclear why more robust statistical analyses were not conducted (e.g., MANOVAs, correlations, or even mediation analyses). Could you please shed some light on this? Additionally, it is necessary to indicate which statistical software was used and its version.
Reply: We added the requested information in the Statistical analyses section. Regarding your last point, we don’t have hypotheses that would require a mediation analysis or multiple dependent variables; however, we agree that correlations could be of interest, so we included a correlation plot in the supplementary materials with reference in the text (page 10, lines 6-8), thank you.
- Lastly, in relation to the Method section, the absence of a brief description of the Procedure stands out (e.g., where and by whom was the evaluation conducted? How many sessions? How many hours per session? Did the sessions include breaks? What was done in each of the sessions?
Reply: Instruments were divided into two batteries, the first consisting of TAQ - STAI - BDI - IPQ; the second of FSCRS - CPAQ - FIQ - DPES. The estimated time for each battery was between 20 and 30 minutes. One or two days passed between one battery and another. Furthermore, the examiners explained to the participants that they could make pauses or stop the compilation at any time. (the evaluation was done in the presence of an examiner trained in this activity). Page 9 (lines 30-33).
- Regarding what is specified in the Discussion, I believe that the relationship between the clinical characteristics of the patients and the possible occurrence of psychosis needs a more in-depth qualitative analysis, as this was introduced by the authors as a novel aspect of the study; however, the information derived from the analyses conducted is too limited. The relationship between them is not clearly explained.
Reply: These analyses of this work are based on the explicit dual aim of outlining the psychological profiles of fibromyalgia by considering the risk of developing psychotic symptoms. We agree on the necessity to further discuss this point (page 16, lines 5-24).
- In conclusion, it would be interesting for the authors to further explore how the observed findings could directly influence the psychological treatment of the participants
Reply: Thank you. ACT-based interventions were hypothesized to have a greater effect, but this is an ongoing investigation. Understanding what the most common symptoms, traits, and behaviors are, it could be easier to help patients reduce their symptoms and improve their quality of life. We addressed further this point in conclusion.
We thank the reviewer for the contribution to our paper.

Round 2
Reviewer 1 Report
Comments and Suggestions for Authors
The article has improved significantly. It was difficult to follow the explanations of the changes as the page and line references do not correspond to the 2nd manuscript submitted. It was also difficult to identify the new references as they are not highlighted in yellow in the reference list. The rate of plagiarism remains high (37%). The percentage of updated references remains low (41%). Reference 30 does not correspond to the current 44, it is another article and another year. 50% of the references should be from 2020-2024. 14 references are included, but they are old (9,12,14,15,16,17...).
Author Response
REVIEWER 1 – ROUND 2
The article has improved significantly. It was difficult to follow the explanations of the changes as the page and line references do not correspond to the 2nd manuscript submitted.
Reply: we have written the previous rebuttal letter referring to the pages and lines as we saw them in the document, for its visualization. There may have been intermediate changes on the manuscript. We have numbered the lines also in this version to hopefully make it easier to find the changes made. We hope that is clear now.
It was also difficult to identify the new references as they are not highlighted in yellow in the reference list.
Reply: We have not highlighted the references in the previous version, apologies. We did it now in this version.
The rate of plagiarism remains high (37%). The percentage of updated references remains low (41%).
Reply: we checked that the paper does not present any form of plagiarism. We used different tools (see this example below) and our paper results as original. However, in case the reviewer still finds plagiarism, we would ask to share the source where such detection is scanned. We hope that this aspect is ok now.
Reference 30 does not correspond to the current 44, it is another article and another year.
Reply: apologies but we do not understand this comment.
The reference list has been completely revised.
50% of the references should be from 2020-2024. 14 references are included, but they are old.
Reply: We measured the percentage of cited works, and it seems adequate now to us (50,8%), thanks.

Round 3
Reviewer 1 Report
Comments and Suggestions for Authors
The article has improved significantly. It was difficult to follow the explanations of the changes as the page and line references do not correspond to the 2nd manuscript submitted.
Reply: we have written the previous rebuttal letter referring to the pages and lines as we saw them in the document, for its visualization. There may have been intermediate changes on the manuscript. We have numbered the lines also in this version to hopefully make it easier to find the changes made. We hope that is clear now.
It was also difficult to identify the new references as they are not highlighted in yellow in the reference list.
Reply: We have not highlighted the references in the previous version, apologies. We did it now in this version.
The new version also fails to highlight the new references. They should have been in red, the yellow does not appear.
The rate of plagiarism remains high (37%). The percentage of updated references remains low (41%).
Reply: we checked that the paper does not present any form of plagiarism. We used different tools (see this example below) and our paper results as original. However, in case the reviewer still finds plagiarism, we would ask to share the source where such detection is scanned. We hope that this aspect is ok now.
Allí sigue siendo un 33% de plagio, te envié una captura de pantalla para que la compruebes. El 6% es plagio con la tesis del autor principal. El plagio se ha reducido del 12% en el primera versión al 6% en esta última versión. Turnitin se utilizó como herramienta para Comprueba si hay plagio.
La referencia 30 no corresponde a la 44 actual, es otra artículo y un año más.
Reply: apologies but we do not understand this comment.
The reference list has been completely revised.
Reference 30 in the first version: M. Passarelli, L. Casetta, L. Rizzi and R. Perrella, was not reference 44 in the second version, it now corresponds to reference 43.
50% of the references should be from 2020-2024. 14 references are included, but they are old.
Reply: We measured the percentage of cited works, and it seems adequate now to us (50,8%), thanks.
Now the references are more than 50%, which is correct.
